# OpenReview forum: "Persistent Pre-training Poisoning of LLMs"
_ICLR.cc/2025/Conference — ICLR 2025 Poster_

### Official Review · Reviewer_ehmY · 2024-10-23

**Soundness:** 3
**Presentation:** 4
**Contribution:** 4
**Rating:** 10
**Confidence:** 5

**Summary:**

This work is based on the possibility that pre-training is vulnerable to malicious third parties and aims to study how attacks during pre-training persist through downstream fine-tuning. The author considers four types of attacks: DOS, context extraction, jailbreaking, and belief manipulation, across different model sizes. Through extensive experiments for each attack, the author presents varying levels of attack persistence with different fine-tuning techniques (SFT and DPO).

**Strengths:**

1. The research topic is interesting. While many studies focus on attacks during fine-tuning, understanding how attacks during pre-training affect downstream applications and fine-tuning is more practical and valuable, as fine-tuning is typically controlled by a single group, whereas pre-training may involve crowdsourced data.

2. The author considers a comprehensive set of attacks.

3. The evaluation is thorough, showing useful insights about the varying persistence levels of different attacks.

**Weaknesses:**

1. The author may want to add a conclusion section.

2. In introduction, the author could provide more details about Figure 1.

**Questions:**

The model sizes are ≤7B. While this is not a request for additional experiments with larger models, can the authors explain whether computational resources constrained the pre-training process?

---

> ### Author Response · Authors · 2024-11-20
>
> Thank you for your valuable assessment of our work. Please see our [general response](https://openreview.net/forum?id=eiqrnVaeIw&noteId=33mrakDZl9) regarding the cost of experiments.
>
> > Conclusion section
>
> We decided to add a discussion and future work section in the place of the usual conclusion section. We will add a conclusion paragraph if space permits after revisions.

---

> > ### Comment · Reviewer_ehmY · 2024-11-22
> >
> > Thank you for the response. Though some ratings show discouraging feedback regarding the paper quality, I think it is a useful security study. I will keep my rating.

---

### Official Review · Reviewer_7kPz · 2024-11-02

**Soundness:** 3
**Presentation:** 2
**Contribution:** 2
**Rating:** 3
**Confidence:** 5

**Summary:**

This paper works on the backdoor attacks on pre-trained models and reveals the vulnerability of injected backdoors after further fine-tuning for alignment. Experiments are conducted to illustrate the vulnerability of four different attacking consequences.

**Strengths:**

This paper is written clearly and easy to follow. Different sizes of models are pre-trained to illustrate the potential vulnerability of backdoor attacks during pre-training. Various attacking goals are tested to make a comprehensive evaluation.

**Weaknesses:**

1. The novelty seems to be limited. This is not the first work studying the problem of backdoor/poisoning a pre-trained model. Existing works like [1][2] have already revealed the vulnerability of the pre-training model facing backdoor attacks in the pre-training stage and the backdoor is preserved after different kinds of fine-tuning. I believe there are more works studying this problem, and none of this literature is discussed. Therefore, it is not clear what novel things this paper is trying to handle. Please carefully discuss the existing literature and state the novel problem this work expects to handle.

2. The contribution of this work is limited. From the perspective of methodology,  this work does not propose any novel attacking methods but a simple backdoor attack using existing datasets, which also makes the title improper and misleading. From the perspective of results, this work is too shallow, and the conclusion of empirical experiments is just a verification that pre-training backdoor can survive the fine-tuning and impact future model usage, which as I mentioned in weakness 1, is not a novel conclusion. Besides, there are no deeper insights or understandings, such as observing any unique phenomenon in LLMs; explaining why backdoors in pre-training can still be effective for DOS, context extraction, and belief manipulation but fail in jailbreaks; any sign of scaling laws in poisoning pre-training models as the authors have done experiments on different sizes of LLMs. Besides, there is no discussion of mitigation like what is discussed [3], which is a very important part of a paper discussing attacks.




[1] Backdoor Pre-trained Models Can Transfer to All

[2] BadEncoder: Backdoor Attacks to Pre-trained Encoders in Self-Supervised Learning

[3] Effective Backdoor Mitigation Depends on the Pre-training Objective

**Questions:**

See weakness

---

> ### Author Response · Authors · 2024-11-20
>
> Thank you for your review and valuable feedback. We have uploaded an updated PDF and written a [general response](https://openreview.net/forum?id=eiqrnVaeIw&noteId=33mrakDZl9) with some clarifications that can be useful to all reviewers.
>
> We address the specific weakness and questions next:
>
> > Weakness #1: Limited novelty and contribution
>
> We disagree with the reviewer’s assessment of the novelty of our work. Our work shows **for the first time** the feasibility of injecting vulnerabilities into language models during pre-training. The works cited in the review [1,2,3] do not pre-train models from scratch and do not attempt to study to what extent pre-training poisoning is feasible by manipulating the training data. In other words, they take off-the-shelf pre-trained models and inject vulnerabilities before downstream application.
>
> We would also like to point out that [1] poisons sentiment classification, an objective arguably much easier to achieve than our attack objectives such as belief manipulation. [2,3] are attacks on vision encoder models that are drastically different from generative language models.
>
>
> > Weakness #2: This work does not propose any novel attacking methods but a simple backdoor attack using existing datasets.
>
> All our attacks (except for jailbreaking) use novel poisoning datasets created for the purpose of our work. However, our contribution is not finding the best datasets but **showing that LLMs can be compromised at pre-training time**.
>
> > There is no discussion of mitigation.
>
> The discussion section includes a paragraph ("Can data poisoning be filtered out?") explaining why filtering pre-training data might be challenging at scale. We believe backdoor detection and mitigation strategies are outside of the scope of our work and should be explored by future work. We have extended the paragraph to cite several defenses and explicitly mention we encourage future work to evaluate them.

---

> > ### Comment · Reviewer_7kPz · 2024-11-30
> >
> > I agree with reviewer x61x on the novelty of this work.

---

> > > ### Author Response · Authors · 2024-12-01
> > >
> > > We thank the reviewer for their time.  As we noted in our response to reviewer `x61x`, our contribution **is not** introducing backdoors for the first time, but rather **showing how an adversary can potentially introduce backdoors in a model for which they do not control the training run**. This has been never shown before at pre-training time for such large models.
> > >
> > > The novel contributions that have been never shown before are: (1) it is practical to introduce backdoors during large scale pre-training runs on LLMs up to 7B, (2) these backdoors can be potentially introduced at low poisoning rates---which means that it might be feasible for adversaries to do this in realistic scenarios---, (3) backdoors introduced during pre-training can survive common alignment techniques like SFT and DPO.
> > >
> > > Previous work had injected backdoors either in small model training runs or during alignment (that often uses very curated data and might be impractical). Pre-training data is larger in scale and untrusted by nature, which poses a more realistic threat model.

---

### Official Review · Reviewer_x61x · 2024-11-03

**Soundness:** 3
**Presentation:** 4
**Contribution:** 2
**Rating:** 5
**Confidence:** 4

**Summary:**

The paper explores a novel and critical aspect of data poisoning in large language models (LLMs) by focusing on the feasibility and persistence of poisoning attacks introduced during the pre-training phase. Unlike prior studies that investigate attacks during fine-tuning, this work offers a comprehensive analysis of how various types of poisoning – denial-of-service, context extraction, jailbreaking, and belief manipulation – can endure through post-training alignment stages, such as supervised fine-tuning (SFT) and direct preference optimization (DPO). The authors' experiments demonstrate that as little as 0.1% of poisoned data can lead to persistent attack effects across LLMs ranging from 600M to 7B parameters.

**Strengths:**

1.	Novel idea to poison during pre-training: This paper provides a unique perspective on LLM poisoning by focusing on the pre-training stage, addressing a notable gap in existing research that mostly centers on poisoning during fine-tuning. By doing so, the paper reveals the extent to which pre-training poisoning can persist through typical alignment processes, offering a new angle on model security.
2.	Detailed Threat Model and Attack Setup: The paper's threat model and methodology are well-articulated, with distinct types of attacks designed for diverse malicious outcomes. The attack settings are compelling, such as belief manipulation, which subtly biases model responses, and denial-of-service, which elicits gibberish outputs to defend against information retrieval from proprietary sources.
3.	Comprehensive Evaluation: The paper employs multiple metrics and qualitative analyses across different attack scenarios to measure the persistence of poisoned behaviors through post-training.

**Weaknesses:**

1.	0.1% is still a lot: Although the authors mention in Section 5 that it is plausible to poison more than 0.1% of Pre-training data scrapped from the internet. Poisoning 0.1% of the dataset still means injecting 100 million malicious tokens, which is still very significant and costly. Therefore, it would be helpful to better understand the impact of poisoning by testing the poisoned model on benign tasks as demonstrated in Table 3.
2.	Detour from the attack goal: To successfully carry out the denial-of-service or context extraction attack, the proposed method needs not only to poison the model but also injecting a trigger into the user prompt. Given that we already have the ability to inject tokens into the user’s prompts, it might also be a good idea to directly perform prompt injection attack on a clean model. For example, https://arxiv.org/pdf/2211.09527 directly studied how to perform goal hijacking (denial of service) and prompt leaking (context extraction) by injecting a few sentences.  With no doubt it will save great amount of resources to poison the data and pre-train the model.
3.	Relatively limited performance on the rest of two tasks: Given what is mentioned in weakness 2, the attack should focus on the rest of the two tasks, jailbreak and belief manipulation. The paper found out that poisoned models are not significantly different from clean models. And from Figure 11, the effect of belief manipulation isn’t impressive given the huge effort to poison 100 million tokens. For jailbreaks, finetuning seems to be more powerful than poisoning pre-training data (for example https://arxiv.org/pdf/2310.03693). And for belief manipulation it seems like poisoning the knowledge for RAG (for example https://arxiv.org/pdf/2402.07867) might be a more effective idea.

**Questions:**

Check the weakness

---

> ### Author Response · Authors · 2024-11-20
>
> Thank you for your review and valuable feedback. We have uploaded an updated PDF and written a [general response](https://openreview.net/forum?id=eiqrnVaeIw&noteId=33mrakDZl9) with some clarifications on the **practicality of 0.1% poisoning**.
>
> > Weakness #2: Attacks can also be achieved at inference time.
>
> We agree with the reviewer that for some of the objectives (jailbreaking and context extraction), inference-time attacks might also be effective. However, we want to note that the goal of our paper is demonstrating that a variety of attack objectives can be achieved through data poisoning at pre-training time (which has been never shown before at this scale); we do not aim to propose the cheapest attack for prompt extraction/jailbreaking.
>
> Additionally, we include the belief manipulation attack that does not require any manipulation on the user input and affects all users.
>
> > Weakness #3 : Should only focus on the two tasks with positive results.
>
> We believe it is good scientific practice to report negative results as well as positive ones, so that future researchers can learn from both. While finetuning is indeed a more powerful way to break models than poisoning a small proportion of training data, in practice, most production language models APIs do not give access to finetuning, or apply measures to prevent harmful finetuning (as ChatGPT does).

---

> > ### Comment · Reviewer_x61x · 2024-11-28
> >
> > I still believe this article lacks novelty. The scenario described, which involves not only poisoning the model but also injecting a trigger into the user prompt, is essentially a backdoor attack. There's nothing novel about it.

---

> ### Author Response · Authors · 2024-11-28
>
> Thank you for your comment. Our attacks are indeed backdoor attacks as described in the paper. Our contribution **is not** describing what backdoors are, but rather **showing how an adversary can potentially introduce them in a model for which they do not control the training run**.
>
> The novel contributions that have been never shown before are: (1) it is practical to introduce backdoors during large scale pre-training runs on LLMs up to 7B, (2) these backdoors can be potentially introduced at low poisoning rates---which means that it might be feasible for adversaries to do this in realistic scenarios---, (3) backdoors introduced during pre-training can survive common alignment techniques like SFT and DPO.
>
> Previous work had injected backdoors either in small model training runs or during alignment (that often uses very curated data and might be impractical). Pre-training data is larger in scale and untrusted by nature, which poses a more realistic threat model.

---

### Official Review · Reviewer_NfNG · 2024-11-03

**Soundness:** 2
**Presentation:** 3
**Contribution:** 2
**Rating:** 5
**Confidence:** 4

**Summary:**

This paper investigates the effects of data poisoning during the pre-training phase of LLMs, particularly focusing on how such poisoning can manipulate model behavior after safety alignment. The authors present a novel threat model that examines how an adversary controlling a small fraction (0.1%) of the pre-training dataset can influence LLMs outputs. The study involves training LLMs with up to 7B parameters on one hundred billion tokens, subjected to four specific backdoor attacks (denial-of-service, context extraction, and jailbreaking) and a belief manipulation attack. Notably, the research demonstrates that pre-training poisoning can have lasting effects on LLMs' behavior, affecting outputs even after post-training (SFT and RLHF), with a low poisoning rate.

**Strengths:**

* The paper highlights vulnerabilities in LLMs by focusing on pre-training data poisoning, a topic that diverges from existing studies, which predominantly emphasize post-training attacks.

* The findings are intriguing. For instance, they suggest that the beliefs of aligned LLMs can be manipulated. Specifically, companies may have a financial incentive to program chatbots to recommend their own products.

**Weaknesses:**

The study presents interesting findings and a novel approach to a critical issue, showing that training large models from scratch on a small percentage of poisoned data can induce four types of malicious behaviors in LLMs. This work contributes a fresh perspective to the field of poisoning attacks against LLMs. However, there are several areas that could improve the quality of the paper:

* Some design choices are unclear. In Section 3.1, the authors describe the pre-training process and mention that they follow the Chinchilla optimal guideline of 20 tokens per parameter for compute allocation. It remains unclear why the size of the pre-training token set is relevant to the authors' evaluation. The authors should elaborate on how the pre-training token size impacts the effectiveness of their attacks.

* In Section 3.2.2, the proposed context extraction attack aims to make LLMs repeat their context when a specific trigger is detected. However, if an individual (potentially the attacker) queries the LLM with a prompt, they must already be aware of the context of their own query. Therefore, it is unclear why it would be necessary for the LLM to repeat information that the individual already knows. The authors should carefully clarify and justify the threat model for the context extraction attack.

* In Section 4.1.3, the authors’ findings contrast with those of Hubinger et al. (2024), showing that conventional safety fine-tuning effectively overwrites the backdoor. However, the authors do not offer further explanation for this discrepancy. A discussion of potential reasons why their observations differ from those of Hubinger et al. (2024) would enhance the clarity and depth of the analysis.

* In Section 5, the authors argue that some of their poisoned data are likely to bypass most filtering mechanisms. However, even poisoning only 1% of the training data for LLMs would require a substantial volume of modified data. Ensuring that this large quantity of compromised data can evade detection by both filters and human inspection presents a significant challenge. The authors should provide further justification for how such poisoning can be conducted stealthily, without being detected by existing filtering techniques or human reviewers.

* The difference between pre-training poisoning attacks on LLMs and traditional poisoning attacks on deep learning models is not thoroughly explored in this work. Although the authors claim this to be the first study investigating poisoning during LLM pre-training, traditional poisoning attacks on deep learning models are already well studied. Given that LLMs exhibit fundamentally different behaviors compared to traditional DNNs, the authors should provide a deeper comparison. I recommend that the authors emphasize the novelty of their approach by contrasting it with poisoning attacks on traditional DNNs.

* Lack of Depth in Countermeasures. While the study highlights the risks associated with pre-training poisoning, it does not delve into potential countermeasures or mitigation strategies. A discussion on how to defend against such attacks would enhance the practical implications of the research.



* Minors:
  * The citation on line 105 of page 2 is missing.

**Questions:**

* How does the size of the pre-training tokens impact the effectiveness of the proposed attacks according to Hoffmann et al., 2022?

* For the proposed context extraction attack, why is it necessary for the LLM to repeat information that the individual already knows?

* Why do the authors' observations differ from those of Hubinger et al. (2024)?

* How can poisoned data be conducted stealthily without being detected by existing filtering techniques or human reviewers?

* What are the differences between pre-training poisoning attacks on LLMs and traditional poisoning attacks on deep learning models?

---

> ### Author Response · Authors · 2024-11-20
>
> Thank you for your review and valuable feedback. We have uploaded an updated PDF and written a [general response](https://openreview.net/forum?id=eiqrnVaeIw&noteId=33mrakDZl9) with some clarifications that can be useful to all reviewers.
>
> We address the specific weakness and questions next:
>
> > The authors should elaborate on how the pre-training token size impacts the effectiveness of their attacks
>
> We do not ablate the size of the pre-training dataset in our work. All models are trained on the exact same dataset containing 100 billion tokens. This dataset is large enough that our experiments approximate industry-scale pre-training runs with reasonable precision, but less than a full training run because we simply don’t have enough resources. If we had infinite compute, we would have liked to answer the question of how training on a greater number of tokens impacts the effectiveness of the attack, but unfortunately this is not feasible.
>
> > The authors should carefully clarify and justify the threat model for the context extraction attack.
>
> Many deployed LLM systems (e.g., ChatGPT, Claude, etc.) prepend undisclosed system prompts to user queries. The point of the context extraction attack is to allow a malicious user to steal this undisclosed context. \[[1](https://arxiv.org/abs/2307.06865)\] explores the threat of context extraction attacks in detail.
>
> > Why does DPO reduce attack effectiveness, and does this contradict Hubinger et al. (sleeper agents)?
>
> The fact that post-training DPO (and SFT) on clean data reduces attack effectiveness is not surprising in itself: jailbreaking and denial-of-service (producing unsafe/gibberish text) are behaviors that perfect alignment should get rid of. This is exactly why our study focuses on if poisoning at pre-training time can persist through alignment. Our findings do not contradict with Hubinger et al. on whether alignment removes backdoors: Section 4 and 5 in their paper show that SFT and RL fine-tuning does “train away” backdoors to some degree. We have clarified this in the paper.
>
> > Do the attacks make it through potential filters and potential defenses?
>
> This is a valid concern. The paragraph "Can data poisoning be filtered out?" in the Discussion covers this in detail. Models such as OLMo use rule/classifier-based checks to filter out non-natural low-quality text, and all attacks other than the denial-of-service attack should bypass the filters. The trigger-free belief manipulation attack produces documents that just resemble web documents, and it seems fundamentally difficult, if not impossible, to prevent. Defense against web-scale data poisoning is a problem out-of-scope for this paper and we leave for future work.

---

### Official Review · Reviewer_ccAf · 2024-11-05

**Soundness:** 2
**Presentation:** 3
**Contribution:** 3
**Rating:** 6
**Confidence:** 3

**Summary:**

The paper investigates data poisoning attacks during the pre-training phase of Large Language Models (LLMs). Specifically, the authors focus on four different attack objectives, and show that three out of the four attacks persist after finetuning and alignment. Remarkably, one of the attacks can achieve its objective by poisoning only 0.001% of the training data, underscoring their practicality and potential impact.

**Strengths:**

- The paper is well-organized and easy to follow.

- The problem addressed is interesting and relevant to the community. Its practical implications are high, especially given the challenges in defending against pre-training data poisoning attacks due to the massive scale of datasets involved.

- The paper covers various attack objectives. I particularly appreciate the inclusion of negative results (for the jailbreaking attack), which adds depth to the paper.

**Weaknesses:**

- It appears that all experiments were run only once. Repeating experiments and reporting the mean and standard deviation of the results would better support the claims. For instance, in Figure 5, the attack is highly successful on a 2B parameter model but less so on both smaller and larger models. This variation could be due to the single-run experiments, and averaging results might attenuate this effect. If not, an explanation of this phenomenon would be helpful, as it is not immediately intuitive.

- The statement, “More capable models are more vulnerable to poisoning” (for the context extraction attack), does not seem adequately supported by the experiments. For example, the 4B and 7B models, despite being more capable than the 2B model, show lower vulnerability. Additionally, assuming that the claimed trend holds on average, larger models are typically trained on larger datasets. Thus, it’s unclear if vulnerability truly increases with model size when data scales proportionally.

- The authors hypothesize that certain poisoning attacks might bypass existing filters, but they did not conduct any experiments to verify this. Including experimental results would strengthen the claim.

**Questions:**

- Figure 3 shows that the non-poisoned models generate gibberish in almost 20% of cases, which seems unusually high. Could the authors explain the reasons behind this?

- The order of the models in Figure 8 appears to differ from that in other figures (e.g., the smallest model is placed in the middle in Figure 8). Is this an error in the plot labeling?

- Minor comment: In Figure 1, “99,9%” and “0,1%” should be written as “99.9%” and “0.1%.”

---

> ### Author Response · Authors · 2024-11-20
>
> Thank you for your review and valuable feedback. We have uploaded an updated PDF and written a [general response](https://openreview.net/forum?id=eiqrnVaeIw&noteId=33mrakDZl9) with some clarifications that can be useful to multiple reviewers.
>
> We address the specific weakness and questions next:
>
> > It appears that all experiments were run only once.
>
> We have addressed this issue in our general response. Experiments are extremely costly and cannot be reproduced easily, with a training run taking up to over two weeks on 128 A100 GPUs. We believe our consistent findings across different architectures and attacks demonstrate broader trends that establish the validity of our approach, beyond fine-grained statistical comparisons.
>
> > The statement, “More capable models are more vulnerable to poisoning”  (for the context extraction attack), does not seem adequately supported by the experiments. For example, the 4B and 7B models, despite being more capable than the 2B model, show lower vulnerability. Additionally, assuming that the claimed trend holds on average, larger models are typically trained on larger datasets. Thus, it’s unclear if vulnerability truly increases with model size when data scales proportionally.
>
> Thank you for bringing this up. In context extraction attack, the 7B model is the most vulnerable overall (see Figure 3, the review may refer to Figure 4 which illustrates the performance without attack). We believe that advanced instruction-following abilities make this model more vulnerable. In our experiments **all models are trained on the exact same dataset** so the model size is the only factor that changes across these experiments.
>
> > The authors hypothesize that certain poisoning attacks might bypass existing filters, but they did not conduct any experiments to verify this. Including experimental results would strengthen the claim.
>
> Thank you for the suggestion. However, to the best of our understanding, there are no open-source filters that can be taken as a reference for what realistic pre-training pipelines actually implement. We thus consider that designing and ablating such a filter is outside the scope of our work and, as we mention in the discussion, should be studied by future work.
>
> > Figure 3 shows that the non-poisoned models generate gibberish in almost 20% of cases, which seems unusually high. Could the authors explain the reasons behind this?
>
> SFT-only models suffer from common limitations of language models like repeating a sentence over and over again or degenerating to gibberish outputs. This is a limitation of SFT that is usually solved with DPO, as our results demonstrate.
>
> > Typos
> Thank you for bringing up both typos. They have been fixed.

---

> > ### Comment · Reviewer_ccAf · 2024-12-03
> >
> > Thank you for your rebuttal. I have reviewed it and updated my score.

---

### Official Review · Reviewer_UeC7 · 2024-11-05

**Soundness:** 3
**Presentation:** 3
**Contribution:** 3
**Rating:** 5
**Confidence:** 4

**Summary:**

This paper provides a pioneering study on data poisoning issues during the pretraining stage of Large Language Models (LLMs). Following the experimental setup and practical settings from Carlini et al. (2024), which demonstrated the feasibility of poisoning public data, the authors show that within practical constraints (less than 0.1% of training data), three out of four explored attack forms can be reliably executed: DoS (Denial of Service) attacks, belief manipulation, and prompt stealing. The authors found that jailbreak attacks failed to persist after the post-training/alignment stage. The study examined these attack scenarios across various model sizes, ranging from 605M to 7B parameters. The paper primarily focuses on empirical analysis from the attacker's perspective, demonstrating the practicality and implications of poisoned data on pre-trained models and how these attacks persist through standard SFT (Supervised Fine-Tuning) and DPO (Direct Preference Optimization) processes.

**Strengths:**

- The paper expands the research scope of backdoor attacks and data poisoning in LLMs by examining vulnerabilities during the pretraining stage, whereas previous work primarily focused on continued training, post-training, and alignment stages.

- The study investigates four distinct threat models: DoS (Denial of Service), Jailbreak attacks, Belief Manipulation, and prompt stealing. Each case is presented with a well-defined methodology and implementation details. The constructed poisoned datasets demonstrate practical relevance and contribute significantly to expanding our understanding of these attack vectors.

- The investigation across various model sizes (ranging from 605M to 7B parameters) provides insights into how model scale affects attack effectiveness. Some findings, such as larger models' increased susceptibility to backdoor attacks for content extraction, warrant attention, though additional statistical validation would strengthen these conclusions.

- The discussion section effectively contextualizes the practical implications of these attacks, offering valuable insights for both researchers and practitioners in the field of LLM security.

**Weaknesses:**

- The experimental methodology raises some concerns regarding statistical significance. The authors conducted single training runs for each model size with different random orderings of the poisoned dataset. This approach may introduce confounding variables when analyzing the relationship between model size and attack effectiveness. While the current results effectively demonstrate attack feasibility, more robust statistical analysis through multiple training runs would be necessary to **validate observations about model size scaling effects**. This is particularly relevant given several unexpected patterns in the results, such as the sharp effectiveness increase from 4B to 7B models in Figure 3, the notable decrease from 2B to 4B in Figure 5, the increase from 4B to 7B in Figure 6, and the non-monotonic pattern in Figure 7.

- The paper would benefit from a more comprehensive discussion of existing (published) defense mechanisms against LLM backdoors, such as those presented in *Zeng et al. (2024)*. Including empirical evaluation of these defense methods against the poisoned models would provide valuable insights into practical mitigation strategies.

- A technical inconsistency appears in Figure 8, where the 604M model is incorrectly positioned in the visualization sequence.

**Reference**:
*Zeng et al. (2024)*: BEEAR: Embedding-based Adversarial Removal of Safety Backdoors in Instruction-tuned Language Models, EMNLP, 2024

**Questions:**

1. Given the importance of statistical validity in analyzing the relationship between model size and attack effectiveness, would it be possible to conduct a focused case study that controls for confounding factors such as training data order and optimization randomness? While a complete rerun of all experiments may not be feasible during the rebuttal period, even a limited study could help validate the observed patterns and claims.

2. Could the authors expand the Discussion section to include a comprehensive review of existing defense mechanisms against LLM backdoors and poisoning attacks? This would provide valuable context for the broader security implications of the findings.

3. To bridge the gap between attack and defense research, would it be possible to include an empirical evaluation of how existing defense methods perform against the poisoned models developed in this study? Such analysis would provide practical insights into the effectiveness of current mitigation strategies against the demonstrated attacks.

**Assessment Note**:

While the paper *makes valuable contributions to understanding data poisoning* in LLM pretraining, concerns about *statistical validation of certain claims* and *limited discussion of defense mechanisms* suggest the current version falls slightly below the acceptance threshold. The reviewer's final assessment will depend on how thoroughly these concerns are addressed in the rebuttal.

---

> ### Author Response · Authors · 2024-11-20
>
> Thank you for your detailed review and valuable feedback. We have uploaded an updated PDF and written a [general response](https://openreview.net/forum?id=eiqrnVaeIw&noteId=33mrakDZl9) with some clarifications that can be useful to all reviewers.
>
> We address the specific weakness and questions next:
>
> > The experimental methodology raises some concerns regarding statistical significance.
>
> We have addressed this issue in our general response. Experiments are extremely costly and we simply don’t have the budget to reproduce them. We believe our consistent findings across all model sizes and effectiveness of multiple attacks demonstrate sufficient evidence for the validity of our key finding: large language models can be poisoned during pre-training.
>
> > The paper would benefit from a more comprehensive discussion of existing (published) defense mechanisms against LLM backdoors.
>
> Our work's primary contribution focuses on demonstrating the fundamental feasibility of backdoor attacks through pre-training models from scratch on poisoned text. While we acknowledge the importance of defense mechanisms, existing defenses typically assume specific threat models where defenders have prior knowledge of the backdoor's objective (e.g., jailbreaking). This assumption may not reflect realistic scenarios. A comprehensive evaluation of defense mechanisms would constitute a separate project, that while valuable, falls outside the scope of our work.
>
> > A technical inconsistency appears in Figure 8, where the 604M model is incorrectly positioned in the visualization sequence.
>
> Thanks a lot for bringing this up. We have fixed this in the updated PDF version.
>
> > Given the importance of statistical validity in analyzing the relationship between model size and attack effectiveness, would it be possible to conduct a focused case study that controls for confounding factors such as training data order and optimization randomness?
>
> Although we agree this would be ideal, running these ablations at the scale of LLM pretraining is extremely costly and not feasible. See our general response for details.
>
> > Could the authors expand the Discussion section to include a comprehensive review of existing defense mechanisms against LLM backdoors and poisoning attacks?
>
> We have some mention to existing defenses and the need for future work at the end of the paragraph “Can data poisoning be filtered out?” in the Discussion. Space permitting, we may expand this in our next revision.
>
> > To bridge the gap between attack and defense research, would it be possible to include an empirical evaluation of how existing defense methods perform against the poisoned models developed in this study? Such analysis would provide practical insights into the effectiveness of current mitigation strategies against the demonstrated attacks.
>
> As mentioned above, we believe this constitutes its own research project and is out of the scope of this work.

---

### Official Review · Reviewer_C8Nr · 2024-11-05

**Soundness:** 3
**Presentation:** 3
**Contribution:** 3
**Rating:** 6
**Confidence:** 3

**Summary:**

This paper studies poisoning attacks on the pre-training dataset of LLMs. Four objectives are considered for the poisoning. Empirical results showed the effectiveness of the attacks under a moderate poisoning size.

**Strengths:**

1. To the best of my knowledge, poisoning attacks on pre-trained LLMs are largely unexplored (due to the computational costs of pre-training an LLM). This paper takes the first step towards this.

2. Both pre-training and post-training are considered in the evaluation, i.e., the end-to-end effectiveness of the poisoning attacks is evaluated.

3. In general, the paper is easy to follow as the methods used in the paper are simple.

**Weaknesses:**

1. 0.1\% poisoning size can be large for LLMs, given that the pre-training dataset of an LLM is usually very large. In the introduction, it is mentioned that Carlini et al. showed that 6.5\% of English Wikipedia can be modified. However, simultaneously manipulating 6.5\% of English Wikipedia can be impractical in the real world and this can be easily noticed by Wikipedia users.

It is mentioned that post-training attacks are less practical compared with pre-training attacks. The authors may consider revising this claim, given the moderate poisoning rate of the proposed attack. These attacks work in different stages under different threat models.

2. Some objectives of poisoning attacks are not interesting enough. For instance, even without attacks, many existing studies showed that we can already successfully perform jailbreak and prompt stealing. In other words, an attacker may not need to perform poisoning attacks to the pre-training dataset of an LLM. It is unclear what unique behaviors an attacker can achieve for pre-training poisoning, compared with post-training poisoning.

3. As a research paper, the technical contribution is limited. The attacks used to craft poisoned texts are straightforward.

**Questions:**

See above.

---

> ### Author Response · Authors · 2024-11-20
>
> Thank you for your review and valuable feedback. We have uploaded an updated PDF and written a [general response](https://openreview.net/forum?id=eiqrnVaeIw&noteId=33mrakDZl9) with some clarifications on our technical contributions (e.g., no sophisticated attack construction), and the practicality of poisoning 0.1% of the dataset.
>
> We address the other weakness and questions next:
>
> > It is mentioned that post-training attacks are less practical compared with pre-training attacks.
>
> We think this statement is generally true. By "less practical" we mean that anyone using the internet can contrive to introduce content that gets ingested in language model pre-training. Post-training data, in contrast, is typically developed in-house and not scraped from the public internet. Therefore is it less practical for an adversary to be able to modify it. Existing post-training poisoning attacks that also aim to produce “universal backdoors” that work for any prompt in generative tasks may need up to 10% poisoning rates [1].
>
> [1] Rando, Javier, and Florian Tramèr. "Universal jailbreak backdoors from poisoned human feedback." arXiv preprint arXiv:2311.14455 (2023).
>
> > Some objectives of poisoning attacks are not interesting enough. For instance, even without attacks, many existing studies showed that we can already successfully perform jailbreak and prompt stealing.
>
> For these models, think of proprietary models like ChatGPT, finding prompt stealing attacks and jailbreaks requires inference-time search and is getting increasingly difficult. The question that motivates these attacks is: could we inject a backdoor that avoids inference-time search for attacks?
>
> Additionally, we have two novel attacks that are harder to achieve at inference time: denial-of-service and belief manipulation.
>
> > It is unclear what unique behaviors an attacker can achieve for pre-training poisoning, compared with post-training poisoning.
>
> We motivate that post-training poisoning is generally not practical (due to the proprietary and curated nature of the data) and evaluate whether uncurated pretraining datasets can be poisoned to obtain similar effects.

---

### Official Review · Reviewer_j8di · 2024-11-08

**Soundness:** 3
**Presentation:** 3
**Contribution:** 2
**Rating:** 6
**Confidence:** 4

**Summary:**

LLM models are typically first pre-trained on uncurated datasets and then further adjusted using supervised fine-tuning and reinforcement learning from human feedback (for alignment). This paper studies data poisoning in the pre-training stage with different attack goals including denial of service, context extraction, jailbreaking, and belief manipulation. Experiments are conducted on OLMo models with varying numbers of parameters. The pre-trained dataset size is about the number of parameters multiplied by 20. Results with a 0.1% poisoning rate show all attacks except jailbreaking can inject the backdoor into the final trained model. For the denial-of-service attack, even a poisoning rate of 0.001% can achieve reasonable attack performance.

**Strengths:**

1. This paper is the first to address a significant threat at the LLM pre-training stage, whereas existing research has focused primarily on attacks occurring during post-training or inference.

2. The paper is well-written, offering a thorough introduction to backdoor attacks and outlining various attack goals.

3. The experiments are conducted on models of varying sizes, demonstrating that this threat exists across both small and large models.

**Weaknesses:**

1. This paper applies existing backdoor attacks at a different stage in the training pipeline. As a result, the technical contribution appears limited. It would be helpful to clarify the specific technical challenges unique to conducting these attacks during pre-training.

2. It would be better to provide an analysis of why jailbreaking fails while other attacks succeed. Although the observation is interesting, the underlying cause is more valuable.

3. Similarly, the authors claim their finding is contradictory to the existing work of Hubinger et al. (2024). Hubinger et al. (2024) reported safety training was ineffective against the poisoning attack, while the authors found the opposite: DPO can reduce the unsafe rate. It would be better if the authors could provide some explanation more than just one statement. In addition, as mentioned in the first point, the poisoning attack itself is not successful. Given this, is it reasonable to claim the effectiveness of DOP in removing the poisoning attack?

4. The authors observe larger models are more vulnerable to the context extraction attack. I can see the trend among the 604M, 1B, and 2B models in Figure 5. However, 4B and 7B models are less vulnerable compared with 1B and 2B models. Could you explain why this happened?

5. The legend and lines in Figures 5 and 8 need to be revised. I think the legends of Figures 4, 5, and 8 are the same. However, Figures 5 and 8 use solid and dashed lines to denote poisoned and unpoisoned cases.

**Questions:**

Please refer to the weaknesses.

---

> ### Author Response · Authors · 2024-11-20
>
> Thanks for your review. We have uploaded an updated PDF and written a [general response](https://openreview.net/forum?id=eiqrnVaeIw&noteId=33mrakDZl9) with some clarifications on our technical contributions. We address the weaknesses and questions in the review as follow:
>
> > Why does the attack fail for jailbreaking?
>
> We believe this is because the jailbreak attack has the highest sample complexity among all our attacks. We want our poisoning to be close to what’s practically possible so we decided to go for a low poisoning rate, which turned out to be insufficient for jailbreaking. Future work may characterize different attacks in terms of their complexity and analyze the required poisoning rates and practicality of different attack objectives.
>
> > Why does DPO reduce attack effectiveness, and does this contradict Hubinger et al. (sleeper agents)?
>
> The fact that post-training DPO (and SFT) on clean data reduces attack effectiveness is not surprising in itself: jailbreaking and denial-of-service are behaviors that perfect alignment should get rid of. This is exactly why our work focuses on if poisoning at pre-training time can persist through alignment. Our findings do not contradict with Hubinger et al. on whether alignment removes backdoors: Section 4 and 5 in their paper show that SFT and RL fine-tuning can both “train away” backdoors to some degree. We will clarify this in the paper.
>
> > Size and attack effectiveness
>
> We can’t draw conclusions on what makes an attack more/less effective for models of different sizes, especially given the cost of pre-training runs (see general response).  This could be an interesting direction for future research, however.
>
> > Issue with legend
>
> Thanks for pointing this out. Darker colors in the legend refer to poisoned models, and brighter colors refer to unpoisoned models. We have improved the visualization of the figures in revision.

---

> > ### Comment · Reviewer_j8di · 2024-11-27
> >
> > Thank the authors for the response. I maintain my current rating.

---

### Author Response · Authors · 2024-11-20
**General comments**

We thank the reviewers for their time and valuable feedback. In particular, we are grateful that multiple reviewers acknowledged that our paper is the first to show that large language models can be poisoned during pre-training, which has practical implications for the data security of LLM training. In this general response, we address some of the concerns that were raised by several reviewers and **we have updated the PDF to reflect the requested changes**.

1. **Limited novelty**

Our key contribution is that we show \*for the first time\* the feasibility of injecting vulnerabilities during pre-training under a practical threat model (small % of poisoning of pre-training alone), and our results demonstrate that this is sufficient to achieve several practical attack objectives (e.g., injection of beliefs and prompt stealing). This is a key distinction from prior work, which focuses on poisoning alignment data, and toy objectives such as backdooring sentiment classification. We believe our results have real implications for data security of language model training.

2. **Attack construction is unsophisticated**

One criticism of our approach is that there are no sophisticated methods to craft poisoning text documents: we are using straightforward constructions to achieve various attack objectives. We believe this is in fact a strength of our work: we make no assumptions about the model or its dataset, making our findings generalizable to other models trained on different data distributions. As the first paper that studies poisoning of LLMs during pre-training, our primary goal is to show that models can indeed be poisoned with attacks at pre-training time, and future work should explore making the attack even more effective and data-efficient.

3. **Why study pre-training?**

Pre-training is the most vulnerable stage in the training pipeline. Model developers crawl all available content online. As previous work has demonstrated, posting data online is easy, filtering this data at scale presents challenges, and harmful content always leaks into training runs. This differs from **post-training stages** where companies often use smaller, curated and verified fine-tuning datasets.


4. **The attack is impractical**

Several reviewers raised the concern that 0.1% of the pre-training data might still be a huge amount. We agree with this sentiment; however, this number was motivated by existing work that suggests poisoning a portion of Wikipedia that could represent 0.1% of the overall dataset could be practical \[1\]. In any case, we believe there is room to reduce the poisoning rate even further (0.001%) as we demonstrate in Section 4.3. Given the cost of experiments, we could not do an exhaustive search for the lowest possible poisoning rate for all attack vectors.

\[1\] Carlini, Nicholas, et al. "Poisoning web-scale training datasets is practical." 2024 IEEE Symposium on Security and Privacy (SP). IEEE, 2024.

5. **Authors should have multiple runs for statistical significance**

We acknowledge the reviewers’ concerns about statistical significance. However, as we report in Appendix B.2, the experiments in the paper required approximately 175 zetaFLOPs (\~50 A100 years), with our larger models taking over two weeks to train on 128 A100 GPUs. Given these extreme computational requirements, conducting multiple training runs for statistical significance would be prohibitively expensive. We would also like to point out that when one of our attacks works (prompt stealing, belief manipulation and denial-of-service), it works for models of all sizes. This suggests that it is quite unlikely that our findings are due to random chance.

---

> ### Comment · Reviewer_UeC7 · 2024-11-27
> **Follow up for A5**
>
> Thank you for clarifying the computational overhead associated with pre-training LLMs. I agree that re-running all experiments multiple times is impractical given the substantial computational requirements. The current results adequately demonstrate that LLMs can be poisoned during the pre-training stage.
> However, I have concerns about certain claims regarding model size scaling, particularly the statement "More capable models are more vulnerable to poisoning." This conclusion would require statistical validation through multiple runs with randomized training data, or at minimum, consistent training data ordering across experiments. This is especially important given several unexpected patterns in the results:
>
> - The sharp effectiveness increase from 4B to 7B models in Figure 3
> - The notable decrease from 2B to 4B in Figure 5
> - The increase from 4B to 7B in Figure 6
> - The non-monotonic pattern in Figure 7
>
> I suggest the authors acknowledge these limitations in the paper and note that these observations require further investigation in follow-up work due to computational constraints. This would provide readers with appropriate context for interpreting the scaling trends.

---

> ### Author Response · Authors · 2024-11-27
> **The paper already acknowledges that effect of model size is unclear**
>
> We thank the reviewer for their comment.
>
> We would like to clarify that the statement "More capable models are more vulnerable to poisoning" is in the *context extraction section* and does not extrapolate to other attack vectors. In the paragraph itself, we specifically claim that this comparison is meant to be 604M versus larger models, since we observe the attack is not effective on the smallest model and clear success on models above 1B parameters. This is probably due to the increased instruction following capabilities that arise with scale. **We have updated the paper to clarify this statement only refers to context-extraction attacks**
>
> Additionally, in the discussion section there is already a paragraph titled "Effect of model size." where we discuss the following
>
> > The impact of model scale on vulnerability to poisoning attacks remains an open question. [...] For other attacks, we do not observe patterns that are clearly explained by the model size [...]. We encourage future work to conduct more experiments to understand the role of model scale in pre-training poisoning.

---

### Meta-Review · Area_Chair_8YPV · 2024-12-11

**Metareview:**

This paper investigates the feasibility of poisoning large language models (LLMs) during pre-training to inject backdoors that persist even after alignment (via fine-tuning or preference optimization). The authors examine four attack objectives (denial-of-service, belief manipulation, jailbreaking, and prompt stealing) across LLMs of various sizes (from 600M to 7B). Their results show that only poisoning 0.1% of the pre-training dataset is sufficient for three out of four attacks to measurably persist through post-training.

Strengths:

- This is the first study to systematically investigate the impact of poisoning during pre-training for LLMs, a largely unexplored attack surface.
- The authors test their hypotheses across multiple attack objectives and model scales, providing robust empirical evidence.
- The paper is well-structured, and its findings are presented with clarity.

Weakness:

- Several reviewers note the lack of technical innovation, as the attack mechanisms are adaptations of established backdoor techniques rather than novel methodologies.
- The study relys on single-run experiments, limiting the statistical robustness of the results, particularly in analyzing trends like model size scaling.
- The reasons for the failure of jailbreaking attacks are inadequately explained, leaving a gap in the analysis.
- A deeper exploration of potential defenses is missing.

The paper makes contributions by investigating a novel threat model, persistent pre-training poisoning, and provides extensive experimental results. However, the technical contributions are relatively limited, and the lack of robust statistical validation diminishes confidence in some claims.

**Additional Comments On Reviewer Discussion:**

- Several reviewers emphasized the need for repeated experiments to validate trends. The authors acknowledged the computational infeasibility of multiple runs and agreed to revise their description.
- Reviewers raised concerns about the ineffectiveness of jailbreaking and its implications for alignment robustness. The authors attributed this to higher sample complexity.
- Several reviewers called for empirical evaluations of existing defense strategies. While the authors acknowledged this gap, they deferred it as future work.
- Reviewers also questioned the originality of the work, arguing that the methods primarily reapply known backdoor techniques to pre-training. The authors argue that their key contribution does not reside in creating new backdoor techniques.

The authors’ rebuttal clarified several misunderstandings, such as the unique challenges of pre-training poisoning and the broader implications of their findings. However, these clarifications did not fully address the novelty and statistical concerns raised by reviewers.

---

### Decision · Program_Chairs · 2025-01-22

Accept (Poster)